# Pioneer Tree *Bellucia imperialis* (Melastomataceae) from Central Amazon with Seedlings Highly Dependent on Arbuscular Mycorrhizal Fungi

**DOI:** 10.3390/jof9050540

**Published:** 2023-05-02

**Authors:** Ricardo Aparecido Bento, Cândido Barreto de Novais, Orivaldo José Saggin-Júnior, Luiz Antonio de Oliveira, Paulo de Tarso Barbosa Sampaio

**Affiliations:** 1Postgraduate Program in Tropical Forest Sciences (PPG-CFT) at the National Research Institute of Amazonas (INPA), Federal Institute of Education, Science and Technology of Amazonas (IFAM), Manaus 69086-475, Brazil; 2Department of Soil at the Federal Rural University of Rio de Janeiro (UFRRJ), Seropédica 23897-000, Brazil; candidobnn@yahoo.com.br; 3Laboratory of Mycorrhiza of the Brazilian Agricultural Research Corporation—Embrapa Agrobiology, Seropédica 23891-000, Brazil; orivaldo.saggin@embrapa.br; 4Laboratory of Ecology and Biotechnology of Microorganisms from the Amazon, Tropical Forestry of the National Research Institute of Amazonas (INPA), Manaus 69060-001, Brazil; luiz.oliveira@inpa.gov.br

**Keywords:** inoculants, mycotrophic species, phosphorus, reforestation, seedlings

## Abstract

*Bellucia imperialis* is one of the most abundant pioneer tree species in anthropized areas of the Central Amazon, and has ecological importance for the environmental resilience of phosphorus (P)-depleted areas. Thus, we investigated whether *B. imperialis* depends on symbiosis with arbuscular mycorrhizal fungi (AMF) to grow and establish under the edaphic stresses of low nutrient content and low surface moisture retention capacity of the substrate. We tried three AMF inoculation treatments: (1) CON—no mycorrhizae; (2) MIX—with AMF from pure collection cultures, and (3) NAT—with native AMF, combined with five doses of P via a nutrient solution. All CON treatment seedlings died without AMF, showing the high mycorrhizal dependence of *B. imperialis*. Increasing P doses significantly decreased the leaf area and shoot and root biomass growth for both the NAT and MIX treatments. Increasing P doses did not affect spore number or mycorrhizal colonization, but decreased the diversity of AMF communities. Some species of the AMF community showed plasticity, enabling them to withstand shortages of and excess P. *B. imperialis* was shown to be sensitive to excess P, promiscuous, dependent on AMF, and tolerant of scarce nutritional resources, highlighting the need to inoculate seedlings to reforest impacted areas.

## 1. Introduction

*Bellucia imperialis* Sald. & Cog. (Syn. *Bellucia dichotoma* Cogn) belongs to the Melastomataceae A. Juss family [1], and is one of the most important neotropical pioneer tree species in Central Amazonia due to the high number of individuals per area [2]. The species occurs in practically all mined areas of the Urucu Oil Province (PPU) [3] (municipality of Coari, state of Amazonas, Brazil), which have dystrophic soils that are acidified and naturally depleted in inorganic phosphorus (Pi) [4].

These impacts on the PPU area are aggravated by the removal of the surface layer of the soil, which reduces the biota and, in addition to becoming gaps with compact soils that are subject to erosion [5], these areas have become difficult to reforest through the planting of seedlings [6], which delays anthropic soil cover [7], even for seedlings of the genus *Bellucia* [8]. The disparity between the potential of *Bellucia* and the negative results of the species in the reforestation of the PPU areas [9] allowed us to hypothesize whether *B. imperialis* seedlings depend on symbiosis with arbuscular mycorrhizal fungi (AMF) to establish and grow in gaps with low fertility and depleted biological communities.

AMF (Glomeromycota phylum) colonize the roots of more than 80% of plant species [10], forming arbuscles and coils, which serve as interface structures for the bidirectional transfer of nutrients between symbionts [11]. The plant provides lipids, carbohydrates, and other growth factors to the fungi [12], and, in return, in addition to enhancing nutrition and hydration [13], this protects the plants against biotic and abiotic stresses [14]. Areas impacted by mining modify competitive relationships, affecting plant establishment and diversity in a community [15], as well as plant responses to mycorrhizal fungi [16]. In this way, symbiosis with AMF can reflect positive, neutral, or negative plant growth, according to the genetic combination of both symbionts modulated by the environment [17].

The occurrence of arbuscular mycorrhizae in the Melastomataceae family may be underestimated [18], and information on how much this biological interaction can enable its seedlings to establish themselves in places depleted in P can help to elucidate the failure of reforestation with the planting of seedlings in gaps from the PPU [4]. Both plants and AMF from P-depleted native areas may be more sensitive to high P fertilization and affect arbuscule development, fungal biomass, and rates of root infection of AMF [19]. The exaggerated supply of P in the shoot can cause toxicity to the plants, interrupting the activation of Rubisco and the defense systems of reactive oxygen species [20], which can neutralize or impair the benefits of mycorrhizal symbiosis [21]. In this context, elucidating the behavior of plants and AMF regarding the level of P fertilization in dystrophic soils can reveal the degree of sensitivity of plants to Pi, and at the same time, their Mycorrhizal Dependence (MD).

Native AMF for inoculation in native pioneer plants can be configured in a successful biotechnological package for the restoration of degraded gaps [22], which motivated us to evaluate the growth of *B. imperialis* at five levels of P, with or without inoculation with native AMF from the PPU (NAT) and/or from the Fungi Collection of Embrapa Agrobiology (MIX); in doing so, we aimed to evaluate the MD of *B. imperialis*, its efficiency in responding to applied phosphorus, and the need for the mycorrhizal inoculation of seedlings for planting in environments with depleted soil and biota, such as the mined areas of the PPU.

## 2. Materials and Methods

### 2.1. Experimental Design and Implementation of Treatments

The experiment was set up in a greenhouse at the Brazilian Agricultural Research Corporation—Embrapa Agrobiology, Seropédica-RJ/Brazil (22°45′34.25″ S and 43°40′52.64″ W), in a completely randomized factorial 3 × 5 design, and with three repetitions; this included three AMF inoculation treatments: (1) CON—control not inoculated with AMF; (2) MIX—pure culture soil sample of 11 AMF species from the Fungus Collection of Embrapa Agrobiology; and (3) NAT—a rhizosphere soil sample of *B. imperialis* containing 13 native AMF species. The inoculation treatments were combined with five doses of phosphorus in the nutrient solution (0.0001, 0.1, 1, 10, and 100 µM P), established in a nutrient solution [23] modified to one-tenth of its original strength (Appendix A). Two 750 mL conical plastic pots (10 cm in upper diameter × 6 cm in lower diameter × 15.6 cm in height) were used for each sample unit, with the first pot perforated to drain the solution, filled with 700 mL of sand granulometry between 1 and 2 mm in diameter, and autoclaved at 121 °C, 1 atm, for two cycles of one hour each. The second pot was painted black, not perforated, and used to coat the first pot and retain the nutrient solution next to the substrate. In the first watering, each pot received 450 mL of nutrient solution (65% of the substrate volume), and subsequently, every 15 days, the nutrient solution was drained and replaced with another new 150 mL solution to maintain the supply of salts and the pH of the solution. After 6 months, when the leaves of the seedlings showed signs of chlorosis followed by necrosis, it was decided to harvest the plants.

### 2.2. AMF Inoculants and B. imperialis Seeds

The inoculum of the MIX treatment came from pure cultures of AMF made in *Brachiaria decumbens* (Syn. *Urochloa decumbens*) from the Fungi Collection of Embrapa Agrobiology (COFMEA) of the Johanna Döbereiner Biological Resources Center (CRB-JD). The MIX inoculum dose was 6.5 mL of soil inoculum, containing an average of 20 spores from each of the 11 mixed AMF species, totaling an average of 220 spores applied per plot, in addition to other propagules such as hyphae and colonized roots (Table 1A). The same volume of 6.5 mL of soil inoculum was applied in the NAT treatment, which contained AMF propagules from rhizosphere soil samples of 48 *B. imperialis* trees (Appendix A). When the spores of this NAT dose were quantified, it was found to contain 234 AMF spores or sporocarps and about 13 species identified by spore morphology (Table 1B).

The fresh fruits of *B. imperialis* were processed, and the seeds (0.03 cm in diameter ± 0.01 cm) were dried in the shade. A hole was created in the pots and the following ordwere added: 6.5 mL of washed and autoclaved vermiculite to retain moisture in the upper part of the pot; 6.5 mL of the AMF inoculants (exempt in the CON treatment); 6.5 mL of washed and autoclaved vermiculite; and 0.0152 g of seeds, which equated to an average of 134 seeds (average of 10 repetitions). A water suspension of both of the inoculants used (NAT and MIX) was filtered to exclude all AMF propagules, and was applied to all pots (10 mL/pot) after sowing, aiming to achieve uniformity of the microbiota in the substrate. The seeds germinated for around 15 days after sowing, and there was no thinning of the seedlings.

### 2.3. Evaluation of the Roots and Shoots

After harvesting, the roots were washed in running water, excess moisture was removed with a paper towel, and the roots were weighed to obtain the root fresh mass (RFM), in grams. Soon after, they were submitted to bleaching and pigmentation procedures [24,25], respectively. The mycorrhizal colonization (MC) of the roots was evaluated according to [26], and microscope slides with fragments of colonized roots were mounted and observed under a microscope with a magnification of 400 times. The leaf area (LA) of the plants was measured in cm² using LI-COR Biosciences L1-3100C Area Meter equipment after being weighed, and the shoot fresh mass (SFM) was obtained, in grams. The samples were stored in forced circulation ovens at 60 °C for two days and, after drying, the shoot dry mass (SDM) was measured, in grams. Then, they were macerated in a porcelain mortar for submission to chemical analysis (with the aim of detecting the Shoot Phosphorus Concentration (SPC) in the tissue, in g·kg^−1^) via complexation of the orthophosphate ion (PO_4_^3+^) in the presence of molybdic acid (H_2_MoO_4_), and analyzed using a spectrophotometer [27] at the agricultural chemistry laboratory of Embrapa Agrobiology.

### 2.4. AMF Evaluation

After the substrate of each pot was homogenized, a volume of 50 mL was removed for the extraction of AMF spores through the processes of wet sieving and centrifugation [28,29]. The spores contained therein were quantified under a stereoscopic microscope at 40× magnification and separated by color, size, and shape [30]. Each morphotype was fixed on a slide for microscopy with PVLG (Polyvinyl Lacto Glycerol) and with PVLG + Melzer’s reagent (1:1) to be identified by morphology based on descriptions in an identification manual [31], in original articles on species, and on specialized taxonomy websites (https://invam.ku.edu/species-descriptions, accessed on 15 December 2022). The nomenclature of the species was updated according to the accepted names on the Mycobank (https://www.mycobank.org/page/Simple%20names%20search, accessed on 15 December 2022) and Index Fungorum (https://www.speciesfungorum.org/Names/Names.asp, accessed on 15 December 2022).

### 2.5. Statistical Analysis

The ANOVA procedure was used in the SISVAR program version 5.8 [32] to assess the effects of the the predictive variables (i.e., “Dose” and “Inoc”) and their interactions. For significant relationships (*p* < 0.05), means were compared using the Scott–Knott test [33], regression analyses were performed using TableCurve 3.01 (Jandel Corporation, San Rafael, CA, USA), and graphs were produced using Excel version 2211. RStudio environment (v.2022.12.0) was used to generate a Heatmap graph to account for the AMF community.

## 3. Results

### 3.1. Mycorrhizal Dependence of B. imperialis

*B. imperialis* seedlings that were not inoculated from the CON treatment, after 45 days of seed germination, showed symptoms of shoot growth stagnation, followed by leaf senescence and death. The complete mortality of the seedlings showed that *B. imperialis* has a high need for symbiosis with AMF because, in the treatments where they had been inoculated, there were no such symptoms. This extreme dependence on the AMF mycelial network for seedling survival under the conditions of a substrate that is very poor in organic matter and physical stressful suggests that the dependence of *B. imperialis* on AMF may be mandatory [34].

### 3.2. Effect of Phosphorus on Mycorrhizal B. imperialis Seedlings

The phosphorus dosages in the nutrient solution significantly influenced the shoot fresh mass (SFM), leaf area (LA), shoot dry mass (SDM), root fresh mass (RFM), and shoot P concentration (SPC) (Figure 1), and showed no significant interaction between the predictor variables (i.e., “Dose:Inoc”) (Appendix A).

The leaf area of *B. imperialis* seedlings decreased considerably (*p* < 0.001) from the second dose of phosphorus (0.1 µM) and followed a downward trend with increasing doses of P (Figure 1A). With increasing P dosage, seedlings underwent significantly decreased fresh (*p* < 0.01) (Figure 1B) and dry (*p* < 0.01) (Figure 1C) shoot mass, as well as decreased fresh root biomass (*p* < 0.05) (Figure 1D). In general, these reflections of the decrease in biomass and leaf area can be observed in Appendix A and Figure 1, where it is observed that the *B. imperialis* seedlings clearly showed greater growth in shoot biomass, root biomass, and leaf area at the lowest dose of P (0.0001 µM).

There was a significant effect of P dose on shoot P concentration of *B. imperialis* seedlings (*p* < 0.01), but the regression adjustment was very weak (R² = 0.10). The highest P dose (100 µM) resulted in the highest P concentration in the shoot (Figure 1E).

### 3.3. Effect of AMF Inoculation Treatments on B. imperialis Seedlings

As for the inoculation treatments, the SPC showed a significant difference between treatments, and was 1.38 times higher in the MIX treatment (*p* < 0.05) in relation to the NAT treatment (Figure 2A).

There was no effect of P dose or of the interaction of P dose with inoculation on the variables that quantified symbiosis (mycorrhizal colonization and the number of spores), possibly due to the high coefficient of variation of these variables (Appendix A). However, the effect of the inoculation factor was evident, as shown in Figure 2B,C. The roots of *B. imperialis* from the MIX treatment had higher average mycorrhizal colonization (58.44%) than the NAT treatment (36.75%), and the NAT treatment showed a large dispersion of data, ranging from 4% to 75%. The MIX treatment was less variable, with mycorrhizal colonization values between 48% and 70%. Regarding the number of spores, the NAT treatment stood out over the MIX, presenting a greater number of spores, resulting in 3.19 times more spores than the MIX treatment (*p* < 0.05), but even so, the NAT treatment was the most variable.

### 3.4. Recovered AMF Community

The survey of AMF species showed differences in their richness (number of species) and in the abundance of individuals (number of spores of each species) between the phosphorus concentrations applied. In Figure 3, scaling of the count of AMF spores per species is observed, recovered in each repetition of the NAT and MIX treatments, to visualize the abundance of each AMF species at different dosages of P.

In the NAT treatment, a total of 8359 AMF spores were extracted, representing a total richness of 16 species, of which three morphotypes could not be identified. The remaining 13 species were identified by morphology to at least the genus level. The species with the highest percentages for the occurrence of spores in the samples in relation to the total number of spores were as follows: Unidentified 1 (41.88%); Unidentified 3 (24.11%); and *Acaulospora* sp. (17.20%), followed by *Glomus* sp.2 (9.31%); *Acaulospora* sp.1 (1.63%); *Ambispora leptoticha* (1.59%); *Glomus* sp.1 (1.49%); *Glomus* sp.4 (1.41%); Unidentified 2 (0.88%); *Ambispora* sp. (0.12%); *Acaulospora* sp.2 (0.12%); *Acaulospora* sp.3 (0.09%); *Scutellospora* sp. (0.04%); *Glomus* sp.3 (0.03%); *Acaulospora foveata* (0.03%); and *Gigaspora* sp. (0.02%). Some AMF species were recovered in only one of the P doses, as was the case with *Gigaspora* sp. and *Ambispora* sp. at a dose 0.0001 µM of P; *Scutellospora* sp., *Glomus* sp.3, *Glomus* sp.2, and *Acaulospora* sp.2 at a dose of 0.1 µM of P; and *Acaulospora* sp.3 at a dose of 100 µM of P. The species Unidentified 1 proved to be abundant, dominant, and resilient, that is, it was the species that presented the highest spore count and the highest percentage of occurrence in the samples, and was not affected by the interference of P concentration treatments in the nutrient solution. The lowest AMF diversity was observed at the highest P dose, with only the Unidentified 2, Unidentified 1, and *Acaulospora* sp.3 species occurring in this treatment. The second dose of P (0.1 µM) accounted for the greatest diversity in AMF, with the occurrence of nine species being detected. Of the AMF species inoculated in the NAT treatment (Table 1B), only three were detected after the experiment was conducted, namely *Ambispora* sp., Unidentified 1, and Unidentified 2. The morphological characteristics present in *Acaulospora* sp. were similar to those of *A. mellea*, such as the presence of a scar and the roughness of the secondary spore wall (Figure 4); thus, they were possibly the same species. 

In the MIX treatment, all 11 species of AMF inoculated at the beginning of the experiment were recovered at the end, in at least one repetition of the different phosphorus dosages. A total of 2621 AMF spores were extracted, with more than 86.95% of this value corresponding to the four most abundant species, whose percentages of occurrence in the samples, in relation to the total number of spores, were as follows: *Glomus formosanum* (49.17%); *Acaulospora mellea* (14.04%); *Rhizophagus clarus* (13.27%); and *Gigaspora margarita* (10.45%), followed by the less abundant *Cetraspora pellucida* (4.15%); *Acaulospora foveata* (4.08%); *Claroideoglomus etunicatum* (2.82%); *Scutellospora calospora* (1.14%); *Gigaspora candida* (0.45%); *Acaulospora scrobiculata* (0.30%); and *Glomus* sp. (0.07%). Both *R. clarus* and *A. scrobiculata* were extracted at 0.0001 µM, 10 µM doses of P. *Glomus* sp. were recovered only at the 0.0001 µM P dose, and the other AMF species were recovered at all P dosages. The lowest AMF diversity (six species) was observed at the highest P dose, while the lowest P dose accounted for the highest AMF diversity, with all 11 inoculated species recovered at this dose. Only *A. foveata* occurred in both inoculation treatments, NAT and MIX.

## 4. Discussion

### 4.1. Mycorrhizal Dependence of B. imperialis

*B. imperialis* responded significantly to the AMF inoculants tested in the NAT and MIX treatments for growth, and was extremely dependent on these AMF inoculants to survive under the experimental conditions. This suggests that this tree species may exhibit obligate mycotrophy [35] in the initial phase of its growth, failing to form an adult plant in the absence of AMFs. Due to the mortality of all non-inoculated seedlings, in the present study, it was impossible to estimate the phosphorus response curve of non-mycorrhizal plants to quantify the degree of their mycorrhizal dependence, as estimated in [36], but it is sufficient to presume a high degree of mycotrophy in this species.

The inability to grow demonstrated by *B. imperialis* without establishing mycorrhizae, even with the high availability of P in the substrate, suggests that its genotype is extremely dependent on AMF [37,38]. A plant’s growth response to the presence of AMF indicates its need for symbiosis, but does not indicate its degree of mycorrhizal dependence [35], as the response to the presence of AMF varies according to the environmental or edaphic conditions [39] and to the efficiency of the plant–fungus combination [36]. In the case of *B. imperialis*, it is believed that the low reserve of nutrients and energy [18] in its tiny seeds is the genetic factor that most increases its mycorrhizal dependence; this is because after germination, the rapid association of *B. imperialis* with AMF may offer the plant an ecological advantage for seedling survival in an environment with edaphic stresses such as low nutrient content and low water availability [40]. Therefore, the high degree of mycotrophy in this species (which was essentially verified in the present study in the initial stages of plant development) under the stressful edaphic conditions of a substrate composed only of coarse sand, with low adhesion and capillary rise [41], made it impossible for the seedlings to grow in the CON treatment without the AMF mycelium net. On the contrary, in the treatments inoculated with NAT and MIX, sporulation and mycorrhizal colonization were promoted in the plant, as verified in the study by [42] under the conditions of P acquisition under water stress.

Tropical forest plants highly dependent on mycorrhizae have already been widely described, such as *Senegalia polyphylla* seedlings [43] and those described by the authors of [40,44,45,46]. Therefore, the present study corroborates the importance of the interdependence between AMF and tropical native plants [47], in addition to the adaptability of these symbiotic organisms living in P-depleted environments with nutritional strategies to withstand low P, but not excess P. The symbiosis of both promotes the capability and efficiency of absorbing, using, and metabolizing P, as described for *Tachigali vulgaris*, *Trattinnickia rhoifolia*, *Ochroma pyramidale,* and *Ceiba pentandra* [48], which are other pioneer tree species occurring in Central Amazonia.

### 4.2. Response of B. imperialis to AMF Inoculants and Doses of P

Both of the studied AMF inoculants, NAT and MIX, benefited the establishment of *B. imperialis* seedlings under the conditions of a sandy substrate and without organic matter, where there was certainly nutritional stress, particularly in the most superficial part, where the nutrient solution arrived only via capillaries. However, there was a decrease in growth with increasing doses of phosphorus, indicating that excess phosphorus in the rhizosphere of this forest species restricts its growth. In experiments of mycorrhizal inoculation with doses of P, it is common to see a reduction in growth with an increase in the dose of P in mycorrhizal plants, in comparison to non-mycorrhizal ones, due to the greater expenditure of photosynthates with the fungus [49]. In the present study, the non-mycorrhizal plants died and were not available for this type of comparison. The reduction in plant growth with increasing P dosage is not common. Our data indicate that this reduction in growth did not have a significant effect on mycorrhizal colonization and sporulation. The maintenance of colonization and sporulation in high-phosphorus conditions suggests that the mycorrhizal association was necessary for the survival of the plant, and they may have collaborated to keep the foliar levels of this nutrient relatively stable, at least until the fourth dose of P (10 µM) (Figure 1E).

A similar reduction in the growth of mycorrhizal plants with increasing doses of P was observed in forest species adapted to low-P soils from Jarrah Forest, southwestern Australia [50]. These authors verified the decrease in the growth of mycorrhizal plants with the increase in soil P in six forest species among twelve studied species when nitrogen was not applied. A similar result was also verified for *Eucalyptus marginata* [51] and perennial legumes native to Australia [52]. These last authors found that *Kennedia prostrata* and *Kennedia prorepens* showed more severe P toxicity at high soil P doses, with reduced dry biomass. With no P toxicity and no other limitations that induce the plant to need mycorrhizae, AMF becomes superfluous when fertilization eliminates the limitations, mainly P and N. In this situation, plant growth is suppressed by the demand for C from the fungus and generates symbiosis of a parasitic nature [53,54].

The present study evaluated, for the first time, seedlings of the pioneer tree species *B. imperialis*, and it was not possible to draw conclusions about the parasitic condition of the AMF, since the non-mycorrhizal control died. However, a decrease in growth due to excess P supplied to seedlings was strongly evidenced with the presence of AMF, suggesting P toxicity or extreme competition for the absorption of other nutrients, such as Fe and Zn [55]. This hypothesis is strengthened by the improvement in growth obtained at the lowest P dose (0.0001 µM), where the plant–fungus relationship was also more efficient, resulting in greater diversity in the AMF community. It is believed that *B. imperialis* seedlings, when associated with AMF, managed to survive the initial stresses present in the experiment and, through the mycelium and the transfer of nutrients via fungus, reach the most abundant nutrient solution at the bottom of the pot, balance their internal levels of P, and absorb nutrients inhibited by excess P, and thus, grow at all P doses, particularly the lowest.

The *B. imperialis* seedlings established themselves in all pots that had been inoculated with AMF, but showed visible symptoms of P excess, characterized by foliar necrosis at the highest levels of P. Excessive inorganic P (Pi) accumulation in leaf mesophyll cells disrupts the activation of ribulose-1,5-bisphosphate carboxylase/oxygenase (Rubisco) and reactive oxygen species (ROS) defense systems through phytic acid accumulation in the leaves [20]. According to these authors, the excessive accumulation of Pi causes an increase in the concentration of cytosolic glucose-6-phosphate and activates the synthesis of phytic acid. This causes Zn precipitation, decreasing the activities of superoxide dismutase linked to Cu and Zn (Cu/Zn-SOD) and leading to the accumulation of ROS. Simultaneously, excessive P accumulation also decreases Rubisco Activase (RCA) concentration, decreasing Rubisco activation and consequently limiting photosynthesis, and also promoting ROS accumulation. The accumulation of ROS, enhanced by both routes, triggers chlorosis and leaf necrosis.

### 4.3. Responses of Inoculated AMF Communities to Doses of P

P levels did not significantly affect mycorrhizal colonization and fungal sporulation (Appendix A), despite the reduction in the magnitude of these variables being a common trend in experiments with different P doses [19]. This lack of effect is possibly due to the variability of the data, particularly in inoculation with the AMF community native to the plant (NAT). Although there was no significant effect of P doses on mycorrhizal colonization and sporulation, the doses affected the diversity of AMF species recovered after the experiment had been conducted, with a decrease in the AMF species community with increasing P doses (Figure 3).

The MIX-inoculated community, originating from the Fungi Collection of Embrapa Agrobiology, showed a tendency towards greater homogeneity in the abundance and species richness of AMF along the P application gradient compared to the NAT community (Figure 3). In general, the six most abundant AMF species in the MIX community (*S. calospora*, *G. formosanum*, *G. margarita*, *C. pellucida*, *A. mellea*, and *A. foveata*) showed adaptive plasticity to a wide range of available P, as they were recovered at all doses of P and in almost all repetitions (Figure 3). The NAT community, on the other hand, showed a greater decrease in species richness with increasing phosphorus. This behavior differs from what was observed in the tropical forest of Ecuador, where AMF communities adjusted to moderate additions of phosphorus, changing their species composition in relation to the control, but maintaining stable species richness [56]. In the case of the NAT community, few species observed in the inoculant were recovered after we conducted the experiment with different phosphorus levels (Table 1 and Figure 3).

The cultivation of native AMF communities in a greenhouse generally results in a loss of diversity in relation to the initial soil sample [57,58], as the new climate and soil conditions act as a species filter, allowing for sporulation of only some of the native species, and sometimes enabling the sporulation of species that are not observed in field samples. This occurred in the NAT community, where the spores recovered after the artificial conditions of the greenhouse showed greater variability in both the number of spores and the promotion of root colonization (Figure 2). The MIX community, long kept in the collection as pure cultures in a greenhouse, showed species that are more adapted to excess phosphorus, fertilization, and artificial soils, and to the climatic conditions of a greenhouse. This community promoted less variable sporulation and colonization and little change in diversity with varying phosphorus availability. Furthermore, the MIX synthetic community is easily reproducible, while the NAT community is variable on spatial and temporal scales, among other variables. In this way, the MIX community is currently the most indicated for the inoculation of *B. imperialis* seedlings.

P fertilization levels significantly affect the diversity, structure, composition, and function of AMF communities [59,60]. However, other combinations of factors may also have contributed to the variability in the NAT community, such as the intolerance/tolerance of some AMF species to P [56]; propagules with low inoculum viability [61]; the storage time since field collection [62]; the dormancy or inactivation of propagules of some species [63]; the lack of adaptation to the sand substrate [64]; and the lack of associated microbiota that some AMF species may need [65]. It is important to emphasize that *B. imperialis* was not incompatible in symbiosis with most AMF species; on the contrary, it was quite promiscuous and showed an ability to associate with many species, regardless of the dose of P. Some species stood out in quantity and may be promising for future inoculation studies in this tree species, such as *A. mellea*, *A. foveata*, *C. pellucida*, *G. margarita*, *G. formosanum*, and *S. calospora* from the MIX community, and the morphotype Unidentified 1 from the NAT community. These species showed abundant spores in the rhizosphere and tolerance to P, which occurred at practically all doses of P (Figure 3).

In the NAT community, the Unidentified 3 morphotype showed great sporulation capacity at a dose of 1 µM of P (Figure 3). The morphotypes Unidentified 2 and *Acaulospora* sp.3 were recovered at the highest dose of P, similarly to *Claroideoglomus lamellosum*, which was the dominant species and proved to be insensitive to the presence of abundant phosphorus in a study by [60]. In general, the lowest doses of P (0.0001 and 0.1 µM) provided the most diverse recovery of the native inoculant of AMF (NAT), with eight and nine species identified, respectively. This reinforces the possibility of the inhibition of photosynthesis via phosphorus toxicity [20], which is associated with the already-known negative effect of phosphorus on AMF [19], resulting in a lower richness of AMF species in the rhizosphere. However, it is important to look beyond the biodiversity of AMF species and verify how the species present relate to their functions of nourishing and helping the plant to tolerate stress [66]. Furthermore, AMF species may exhibit phenotype and genotype changes induced by different hosts or phosphate levels [67]. The NAT community is supposedly adapted to the biotic and abiotic conditions of places characterized by the natural occurrence of *B. imperialis*, including their adaptation to other plants with joint occurrence. For this reason, they show greater variability in mycorrhizal colonization, sporulation, and species richness. In the forest, AMF species form extensive networks of hyphae, which colonize different hosts simultaneously, reaching different habitats or soils with different properties, and generating a genetically diverse population [67]. Local selection, as occurred in the present experiment, changes the structure of populations of AMF species, leading to a less diverse community, and reducing the presence of polymorphic genetic markers, as verified by [67].

## 5. Conclusions

Newly germinated seedlings of *B. imperialis* proved to be totally dependent on AMF to survive under the edaphic stress conditions of nutrient scarcity and low moisture retention capacity, regardless of the level of available phosphorus. *B. imperialis,* under the study conditions, proved to be an obligate mycotrophic species, and consequently, its inoculation is highly recommended during sowing for seedling production, and when it is intended to be introduced to environments with strong disturbances in the soil biota, as in the mined areas of the Central Amazon.

*B. imperialis* is promiscuous in terms of its association with different species of AMF; when mycorrhizal, it efficiently tolerates limited nutritional resources, mainly low P availability, and is very sensitive to excess P fertilization, which causes chlorosis, leaf necrosis, and reduced biomass.

Future studies may be directed toward the evaluation of AMF isolates that result in better efficiency of symbiosis with *B. imperialis* to promote its growth, nutrition, and survival in P-depleted soils, to elucidate its degree of mycorrhizal dependence, and also to investigate the physiological mechanisms of phosphorus toxicity presented by this species.

The present study indicates that inoculating the pioneer tree species *B. imperialis* with a community of AMF species collected in the field (which may be similar to the NAT community), or preferably with the AMF community used for the MIX, could be a potential biotechnological strategy for the recovery of degraded areas in environments depleted by P in Central Amazon.

## Figures and Tables

**Figure 1 jof-09-00540-f001:**
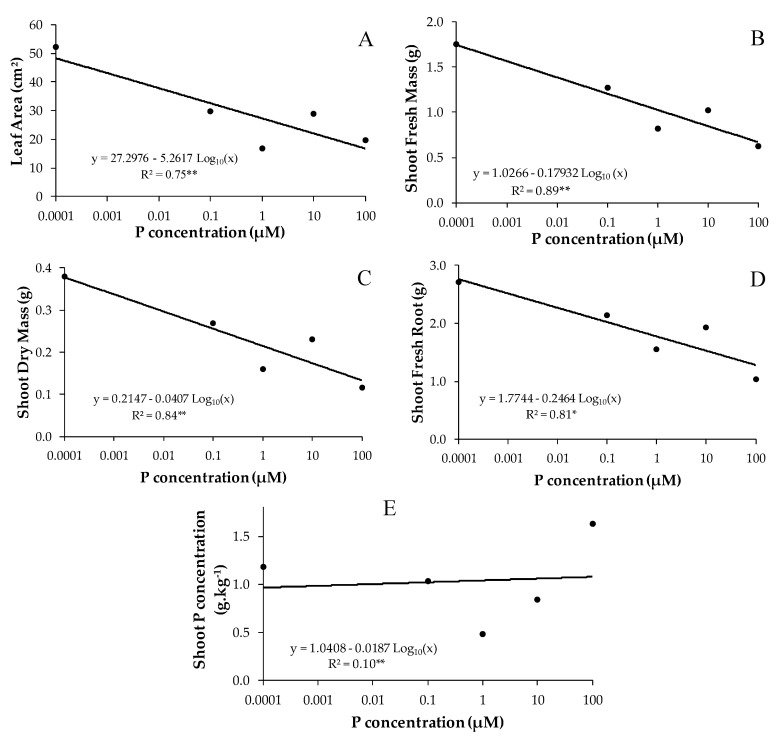
Regression plots between *B. imperialis* seedling growth variables (*y*-axis) and P doses (*x*-axis). Response variables are shoot fresh mass (SFM), leaf area (LA), shoot dry mass (SDM), root fresh mass (RFM), and shoot P concentration (SPC). Bold points plotted in the subfigures (**A**–**E**) refer to the averages obtained from the respective dependent variables in relation to phosphorus doses. In each subfigure (**A**–**E**) is the equation of the straight line and the coefficient of determination (R²) of the respective models. Asterisks refer to the significance level of the F-test of the regression model: * *p* < 0.05; ** *p* < 0.01.

**Figure 2 jof-09-00540-f002:**
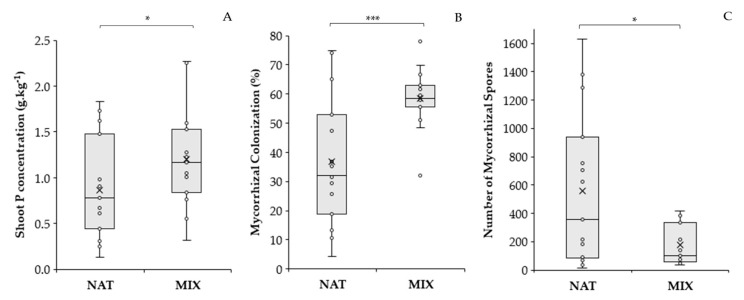
Comparison of arbuscular mycorrhizal fungi (AMF) inoculation treatments NAT and MIX on the variables shoot P concentration (**A**), mycorrhizal colonization (**B**), and AMF spore number (**C**). NAT—inoculation treatment with native AMFs. MIX—inoculation treatment with AMF from the Fungi Collection of Embrapa Agrobiology. In the boxplot, circles represent the observed data; x represents the mean; and the line inside the box is the median. The bars show the range of observed maximum and minimum values. Points outside the bar are outliers. The filled-in boxes show that most of the observations are between the 1st and 3rd quartiles, and the larger the box, the more dispersed the observed data set is. Asterisks refer to the significance level of F in the ANOVA test: * *p* < 0.05; *** *p* < 0.001.

**Figure 3 jof-09-00540-f003:**
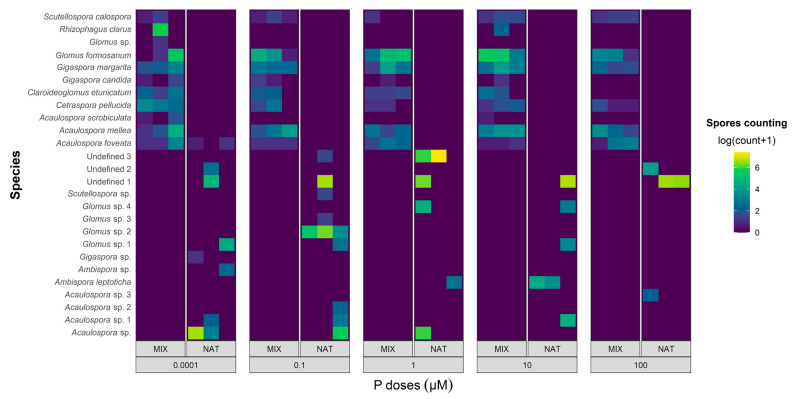
Heatmap of AMF spore count by species in NAT and MIX treatments under different phosphorus dosages, represented as a logarithmic color scale. The values for the number of spores per species in each repetition were Log(count + 1)-transformed and are displayed on the graph according to a color scale: blue for the lowest number of spores and yellow for the highest number of spores. The AMF species are grouped on the graph according to the inoculation treatment, with the first 11 species found in the MIX treatment and the last 16 species detected in the NAT treatment.

**Figure 4 jof-09-00540-f004:**
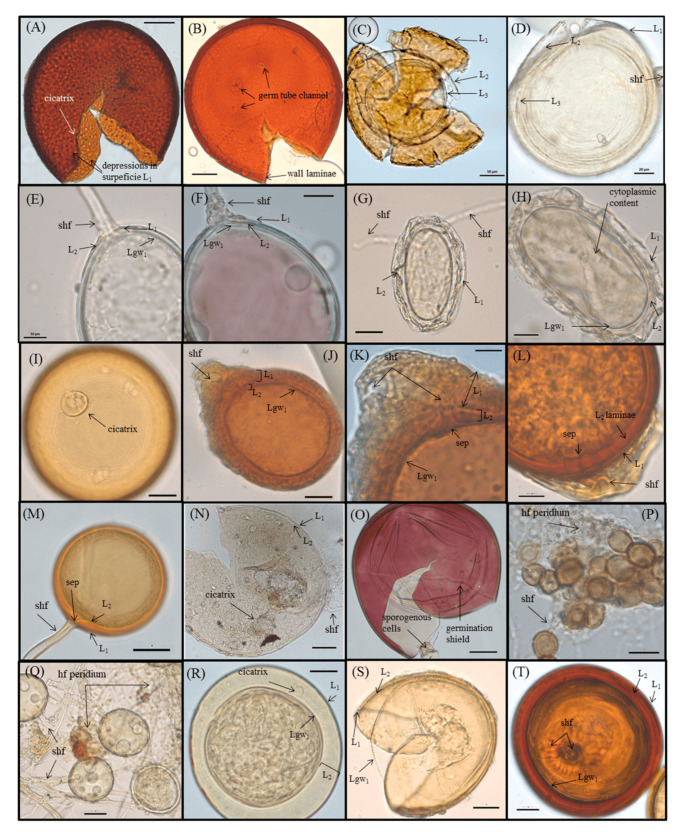
AMF spores recovered from NAT inoculation treatment. (**A**) *Acaulospora foveata*, with the scar and depressions on the surface of the spore wall highlighted (L1). (**B**) *Gigaspora* sp., with detection of laminated layers in the spore wall and germ tube channels of hyphae. (**C**,**D**) *Ambispora leptoticha*, with (**C**) representing an old spore and (**D**) representing a new globoid spore, both with three walls (L1—deep fissures on the surface, visible on the old spore, L2—rounded convex hemispherical protuberances on the inner surface, and L3—compacted concave hemispherical depressions); (**E**,**F**) Unidentified 1, with spore in PVLG preserved in (**E**) and spore with cytoplasm content reactive to Melzer dye (pink color) in (**F**). Both form a sporocarp with dispersed peridium and comprise elongated subtended hyphae. (**G**,**H**) Unidentified 2, with spore wall (L1) hyaline and mucilage, and the thick layer in L2 followed by the germinal wall (Lgw), and with evident cytoplasmic content. (**I**) *Acaulospora* sp., with the scar highlighted and showing similar to the morphological characteristics to *A. mellea*. (**J**,**K**) *Ambispora* sp., with the K spore showing detail of the thick hyphal mantle wall (L1), followed by the continuous layer (L2), with the subtended hyphal wall and germinal wall (Lgw) forming a septum. (**L**) *Glomus* sp.4, showing the presence of subtended hyphae (shf) and mucilaginous wall (L1). Laminated L2 wall and germinal wall form a septum. (**M**) *Glomus* sp.1, with globoid and well-defined L1 and L2spore walls, the latter forming a septum between the subtended hyphae. (**N**) *Acaulospora* sp.3, in which spores are old but with scar and spore walls L1 and L2, as well as the subtended hyphae, well defined. (**O**) *Scutellospora* sp., with spore walls (L1) reactive to Melzer’s stain, and with the sporogenous hyphae and germinal shield highlighted. (**P**) *Glomus* sp.2, with formation in sporocarp and tangle of hyphae forming the peridium. (**Q**) Unidentified 3, with formation in sporocarp, and with hyaline spores globose and connected by well-defined subtended hyphae forming the peridium. (**R**) *Acaulospora* sp.2, with globose spore, thick L2 wall and presence of a scar, similar to the species *A. colombiana*. (**S**) *Acaulospora* sp.1, with globose spore, defined L1 and L2 walls, and a wrinkled germinal wall. (**T**) *Glomus* sp.3, with globoid spore and thick L2 wall. Subtended hyphae are present. Scale bars: 10 µm (**E**,**F**,**H**,**K**,**L**,**P**); 20 µm (**A**,**D**,**G**,**I**,**J**,**M**,**N**,**Q**,**R**,**S**,**T**); 50 µm (**B**,**C**,**O**).

**Table 1 jof-09-00540-t001:** AMF species used as inoculants in the treatments (**A**) MIX—Fungi Collection of Embrapa Agrobiology (COFMEA) of the Johanna Döbereiner Biological Resources Center (CRB-JD), and (**B**) NAT—rhizospheric soil of *B. imperialis* from the native areas of Urucu Oil Province (PPU).

**A. MIX treatment**		
**Species of Arbuscular Mycorrhizal Fungi (AMF)**	**Lineage Code**	**Number of Spores per Dose**
*Acaulospora foveata* Trappe & Janos (1982)	A92	20
*Acaulospora mellea* Spain & N.C. Schenck (1984)	A94	20
*Acaulospora scrobiculata* Trappe (1977)	A38	20
*Cetraspora pellucida* (T.H. Nicolson & N.C. Schenck) Oehl, F.A. Souza & Sieverd (2009)	A70	20
*Claroideoglomus etunicatum* (W.N. Becker & Gerd.) C. Walker & Schuessler (2010)	A44	20
*Gigaspora candida* Bhattacharjee, Mukerji, J.P. Tewari & Skoropad (1982)	A36	20
*Gigaspora margarita* W.N. Becker & I.R. Hall (1976)	A49	20
*Glomus formosanum* C.G. Wu & Z.C. Chen (1986)	A20	20
*Rhizophagus clarus* (T.H. Nicolson & N.C. Schenck) C. Walker & A. Schüβler, 2010	A5	20
*Scutellospora calospora* (T.H. Nicolson & Gerd.) C. Walker & F.E. Sanders (1986)	A80	20
*Glomus* sp.	A100	20
**B. NAT treatment**		
**Species of Arbuscular Mycorrhizal Fungi (AMF)**	**Number of Spores per Dose**	
*Acaulospora mellea* Spain & N.C. Schenck (1984)	29	
*Acaulospora spinosa* C. Walker & Trappe (1981)	4	
*Acaulospora tuberculata* Janos & Trappe (1982)	11	
*Ambispora* sp.	11	
*Claroideoglomus claroideum* (N.C.Schenck & G.S. Sm.) C.Walker & A.Schüßler (2010)	9	
*Glomus glomerulatum* Sieverd. (1987)	19	
*Glomus macrocarpum* Tul. & C. Tul. (1845)	31	
*Glomus multicaule* Gerd. & B.K. Bakshi (1976)	21	
*Glomus* sp.10	9	
*Sclerocystis rubiforme* Gerd. & Trappe (1974)	30	
*Sieverdingia tortuosa* (N.C. Schenck & G.S. Sm.) Blaszk., Niezgoda & B.T. Goto	21	
Unidentified 1	20	
Unidentified 2	19	

## Data Availability

Not applicable.

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
