# Peer review of "Pioneer Tree Bellucia imperialis (Melastomataceae) from Central Amazon with Seedlings Highly Dependent on Arbuscular Mycorrhizal Fungi"

_jof, 2023, doi:10.3390/jof9050540_

Round 1

Reviewer 1 Report

The MS shows originality and scientific relevance. The work shows consistency between the proposed objectives, the applied methodology and the discussion of the results found. I suggest that the authors correct the numbering in the legend of figure 2 as well as in the text of the MS.

Author Response

Dear reviewer,

Thank you for your observation.

Figure numbering in the legend and figure citation number in the manuscript text has been corrected. 

Following the procedure of the MDPI paper, I forward this same comment in attachment. Please see the attachment!

Sincerely,
Ricardo Bento

Reviewer 2 Report

This manuscript was studied the effect of AMF on growth and establish of Bellucia imperialis under different P levels in anthropized areas of the Central Amazon. The results concluded that the growth of B. imperialis dependent on AMF in dystrophic soils. I think the topic is of interest and better understanding of the roles of AMF in reforestation.

Several amendments are suggested as follows:

Line 16 and 47, the initial letters of Arbuscular Mycorrhizal Fungi do not need to be capitalized.

Line 17, please clarify the nutritional and edaphic stress.

There are too many descriptive text in the Introduction section, such as Lines 30-49, please concentrated.

Lines 89-93, I do not know the purpose of using 2 pots.

Table 1, the A and B sections can be combined into 1 table.

Figure 2 caption, what does the 4A, 4B and 4C mean?

Line 204, this is not to explain P effect, but to explain mycorrhizal effect.

Line 207, it should be Figure 2B and C.

Results, it is better to divided into several sub-sections.

In this study, 2 inoculation treatments were conducted, you can discuss and recommend the better one (MIX or NAT) for application in the reforest areas.

There are too many references, especially in the Introduction section.

Author Response

Dear reviewer,

All points raised by your questions were answered following the MDPI script "Response to Reviewer 2 Comments", attached.

Thank you for your input and collaboration for our article.

Yours sincerely,
Ricardo Bento

Reviewer 3 Report

Line 113-114 - Table 1. AMF species used as inoculants in the treatments (1) MIX - Fungi Collection of Embrapa Agrobiology (COFMEA) of the Johanna Döbereiner Biological Resources Center (CRB-JD) and (2) NAT - rhizospheric soil of B. imperialis from the native areas of Oil Province of Urucu (PPU). Numbers (1) (2) must refer to the nomenclature used in table (A) and (B).

Table 1. A. MIX TREATMENT and Figure 3 and line 281-282, line 420: Current names must be used and should be the same in Table 1, Figure 3 and throughout the text.

In Index Fungorum Current names:

Rhizophagus clarus (T.H. Nicolson & N.C. Schenck) C. Walker & A. Schüßler 2010

Cetraspora pellucida (T.H. Nicolson & N.C. Schenck) Oehl, F.A. Souza & Sieverd., Mycotaxon 106: 338 (2009)

Line 122 - … A filtered solution free of AMF propagules… A solution of what? soil?

Line 141-421. Rewrite the sentence. Suggestion: After being homogenized, a volume of 50 mL was removed from the substrate of each vessel for extraction of AMF spores by the process of wet sieving and centrifugation.

Line 188 - The increase in P doses showed a very low positive correlation / showed no correlation?

Line 195-196 – Figure 2. … Shoot P Concentration (4A), Mycorrhizal Colonization (4B), and AMF Spores Number (4C). Should be … Shoot P Concentration (A), Mycorrhizal Colonization (B), and AMF Spores Number (C).

Line 207 - … as shown in Figures 4B and 4C. Should be … as shown in Figures 2B and 2C.

Line 226 – “A. foveata was the only species identified in both treatments.” The sentence is unnecessary as it appears in the results.

Line 229 - 13 were identified by… / 13 species were identified by…

Line 342 - dose of P (10 µM) (Figure 2E). / dose of P (10 µM) (Figure 1E).

Line 392 - … were recovered at all P doses…/… to be recovered (?) at all P doses…

Line 508-697- In References the font and size used is not uniform.

Line 598 - 38. Koske, R.E.; Gemma, J.N.A modified … / 38. Koske, R.E.; Gemma, J.N. A modified …

Line 600 - 39. Grace, C.; Stribley, D.P.A. safer …/ 39. Grace, C.; Stribley, D.P. A safer …

Author Response

(The authors gave the same response as above.)

Round 2

Reviewer 2 Report

The manuscript was revised according to the Reviewers' comments and suggestions. I think this version can be accepted for publication in the journal.